# Undirected C-H Bond Activation in Aluminium Hydrido Enaminonates

**DOI:** 10.3390/molecules28052137

**Published:** 2023-02-24

**Authors:** Chijioke Kingsley Amadi, Ufuk Atamtürk, Andreas Lichtenberg, Aida Raauf, Sanjay Mathur

**Affiliations:** Institute of Inorganic Chemistry, Department of Chemistry, University of Cologne, Greinstr. 6, 50939 Cologne, Germany

**Keywords:** hydride migration, C-H bond activation, aluminium, main group compounds

## Abstract

Two new aluminium hydrido complexes were synthesized by reacting AlH_3_ with the enaminone ligand N-(4,4,4-trifluorobut-1-en-3-on)-6,6,6-trifluoroethylamine (HTFB-TFEA) in different molar ratios to obtain mono- and di-hydrido-aluminium enaminonates. Both air and moisture sensitive compounds could be purified via sublimation under reduced pressure. The spectroscopic analysis and structural motif of the monohydrido compound [H-Al(TFB-TBA)_2_] (**3**) showed a monomeric 5-coordinated Al(III) centre bearing two chelating enaminone units and a terminal hydride ligand. However, the dihydrido compound exhibited a rapid C-H bond activation and C-C bond formation in the resulting compound [(Al-TFB-TBA)-HCH2] (**4a**), which was confirmed by single crystal structural data. The intramolecular hydride shift involving the migration of a hydride ligand from aluminium centre to the alkenyl carbon of the enaminone ligand was probed and verified by multi-nuclear spectral studies (^1^H,^1^H NOESY, 13C, ^19^F, and ^27^Al NMR).

## 1. Introduction

Aluminium hydrides modified with alkoxide [1]; and chelating ligands such as β-ketoiminate ligands offer an interesting class of molecules showing a judicious combination of reactivity and stability (Figure 1) [2]. Such mixed-ligand complexes are of paramount interest in both molecular catalysis and materials synthesis [3,4,5]. Aluminium compounds, especially soluble aluminium hydrides, received wide explorations due to their high reactivity towards protonic reagents and unsaturated compounds containing multiple bonds such as C=O, C=NR, C≡N, and C≡C [6,7]. Recent studies have shown that reactions involving aluminium hydrides usually occurs by deprotonation or hydroalumination, hence a great prospect in main group catalysis. These stoichiometric reactions usually represent the first stage of the complete catalytic cycle. The activation of the substrates and the rejuvenation of the active catalytic molecules depend on the presence of the proper ligands at the core Al-atom [8,9,10]. The type of ligand attached to the aluminium hydride always affects the reaction toward alkenes and alkynes; species with lower coordinate aluminium tend to participate in the hydroalumination process, whilst those with bulky ligands typically go via deprotonation [11,12,13]. Aluminium hydride reactions with R-EH (E=O, S, etc.) and terminal alkynes exhibit outstanding importance in the development of main group catalysts [14,15,16]. Hydride shift triggered C(sp3)–H bond functionalization has recently emerged as a powerful tool for the rapid construction of various useful organic molecules [17,18]. Similar to heavy transition metals, the sp^3^ C-H activation can be induced by an internal hydride shift between a π electron-poor double bond as acceptor and a X-H bond adjacent to an oxygen or nitrogen as a hydride donor [19,20,21].

Herein we report an undirected intramolecular C-H bond activation by the hydride compound [(Al-TFB-TBA)-HCH2] that is able to mediate proton and electron transfer, which suggests new potential applications of aluminium hydride complexes in molecular catalysis. In addition, the new hydrido derivative of the aluminium reported here was used as efficient precursor for the chemical vapor deposition (CVD) of Al_2_O_3_ thin films.

## 2. Results and Discussion

### 2.1. Synthesis of Ligand and Aluminium Compounds 

The enaminone ligand manifests a synergistic combination of reactivity and stability with the fluorocarbon moiety present in the ligand enhancing the volatility by suppressing the intermolecular interaction owing to the increased repulsive forces among the vicinal fluorine atoms [22,23]. The bidentate enaminone ligand **2** was prepared by reacting ethyl vinyl ether with trifluoroacetic anhydride in the presence of pyridine under the elimination of pyridinium trifluoroacetate to obtain 1-ethoxy-4,4,4-trifluorobut-1-en-3-on **1** (ETFB) (Figure 2). The addition of the fluorinated amine in dichloromethane led to N-(4,4,4-trifluorobut-1-en-3-on)-6,6,6-trifluoroethylamine **2** (HTFB-TFEA) under the elimination of EtOH (Figure 2). The freshly prepared enaminone ligand was reacted with in situ synthesized aluminium trihydride (AlH_3_^.^THF) in 1:1 and 1:2 ratios to obtain heteroleptic mono- and di-hydrido-aluminium enaminonate complexes (Figure 3). The facile hydride substitution reaction produced [H-Al(TFB-TBA)_2_] **3** and [(Al-TFB-TBA)-HCH2] **4a** as air and moisture sensitive complexes. The formation of **4a** occurred by rapid hydride shift involving the intermediate [4] (Figure 3), whereas **4a** underwent further hydride migration in solution to give **4b** and **4b′**. The hydride migration from the H-Al-H to the C(sp2)-H bond is demonstrated (Figure 4), revealing the potential role H-Al-H as a proton shuttle and catalyst for the C(sp^3^)–H bond functionalization. The internal redox catalytic reaction caused C(sp3)–H bond functionalization and formation of complexes that are otherwise difficult to synthesise by conventional methods [24,25,26]. Both **3** and **4a** were purified by sublimation to obtain yellow (**3**) and light yellow (**4a**) powders, respectively which is an important prerequisite of CVD precursors.

#### 2.1.1. Structural Characterization of Compounds **3** and **4a**

^1^H NMR spectra measured in deuterated benzene (C_6_D_6_) for structural elucidation of compounds **3** and **4a** confirmed the deprotonation of the ligand by aluminium hydride through the absence of the proton present in the amino group of uncoordinated/protonated ligand that appears at 10.66 ppm (Appendix A). The characteristic peaks for the rest of the ligand backbone were present, however slightly downfield shifted owing to the complexation with the aluminium centre. Despite the clear peaks corresponding to ligand, the hydrides attached to the aluminium centre exhibited significant peak intensity at 3.38 ppm (Appendix A), while the ^19^F showed one singlet at −76.47 ppm that could be assigned to the CF3-group present in the enaminone ligand (Appendix A) for the monohydride compound **3**. This observation corresponds well with similar hydrido complexes reported earlier [27]. 

The ^1^H NMR analysis of **4a** showed two sets of signals with percentages of 73% and 27% respectively (Figure 1), depicting a rigid ligand framework due to complex formation of which two conformational isomers (Figure 5) in solution (**4b** and **4b′**) are present in the ^1^H NMR (and also ^19^F, ^13^C and 2D NMR spectra). This is most obvious from the H3a, H3b and H4 (same for H3a′, H3b′ and H4′) which form a strongly coupled AMX system and differ by the relative orientation of the H3a, H3b vs. H4 protons. Both H3b and H3b′ protons appeared as a doublet of doublets at 2.62 ppm (dd) and 2.74 ppm (dd) respectively, whereas a doublet of triplet may be anticipated for H3a at 3.56 ppm (dt) and H3a′ protons at 3.82 ppm (dt) respectively. Sharp peaks observed at 4.46 and 4.03 ppm could be assigned to hydride groups (Al-H and Al-H′ respectively) and correlates well with previous NMR data on allied derivatives possessing terminal hydrides attached to the aluminium centre [10]. H4 and H4′ protons clearly showed a doublet of doublets of doublets at 5.09 and 4.98 ppm (ddd) respectively, thus there’s stronger coupling between H3a and H4 as seen in the ^1^H, ^1^H NOESY (Figure 2), which allows a guess on the relative orientation of these protons (e.g., equatorial vs. axial or endo vs. exo), therefore one could possibly make a plausible guess here by looking at the crystal structure. 

The ^19^F NMR showed two singlets at −73.84 and −73.99 ppm respectively (Appendix A) which could be assigned to the CF3-group present on the enaminone ligand, which shifted downfield compared to the free ligand (−76.9 ppm). For further characterization of the behaviour in solution of the compounds 4b and 4b′, ^13^C APT spectroscopy allowed the sufficient identification of carbon atoms within the conformation isomers and could be assigned to all carbons of the ligand and are listed in Appendix A. Because most of the proton and carbon chemical shifts has been determined using ^1^H and ^13^C NMR, the exact structural units in 4b and 4b′ were easily determined using ^1^H−^13^C HMQC (Appendix A) and ^1^H−^13^C HSQC (Appendix A) correlations.

The chemical shift in the ^27^Al NMR spectra helps to indicate the coordination number of the aluminium atom. Hexacoordinated aluminium compounds are found at higher fields, whereas four- and five-coordinated aluminium compounds are shifted downfield as broad diffuse signals [28,29]. The solution ^27^Al NMR spectra (Appendix A) of **3** showed broad peaks at 66.4 ppm. This resonance further confirms the 5-cordinated arrangement of 2-fold bidentate ligand and 1-fold hydride ligand at the Al centre [30]; and in agreement with the expected up-field shift with increasing coordination number of the aluminium [31]; whereas **4b** and **4b′** shifted downfield revealing a broad peak at 83.9 ppm (Appendix A) and a weak peak at 191.6 ppm respectively which agrees with reported values of less-coordinated aluminium compounds. Broad peaks observed in alane complexes has been reported to be due to fast relaxation of ^27^Al nuclei which exceeds the coupling constants and makes the observation of ^27^Al-^1^H spin-spin couplings impossible. Therefore, it may be concluded that the broad signals of alane proton as observed in **3** and [(**4b**) (**4b′**)] are as a result of the spin-spin coupling [32].

The FT−IR spectra revealed different vibrations indicating different coordination modes of Al−H hydride ligands as previously reported [33]. Both the ligand and complexes showed significant vibrations in the regions of 3300 cm^−1^−3200 cm^−1^ and 1650 cm^−1^–1450 cm^−1^ (Appendix A), which consist of N−H stretching and bending vibrations, respectively. These peaks are prominently seen in the FTIR spectrum of ligand while completely weak or absent in those of the complexes, indicating the coordination of ligand through the N atom to the Al centre. The existence of the hydride at the aluminium centre could be distinguished from characteristic Al-H stretching vibrations observed at 1676 cm^−1^ (**3**) and at 1914 cm^−1^, 1670 cm^−1^ [(**4b**) (**4b′**)], all in conformation with reported Al-H ranges. Interestingly, the twin peaks observed at 1914 cm^−1^, 1670 cm^−1^ ([(**4b**) (**4b′**)] is a further confirmation of isomers that may be present in solid state. 

Despite several crystallization attempts, single crystals of **3** showed weak diffraction patterns that prevented structural refinement, however the metal-ligand connectivity could be unambiguously established to determine the structure (Figure 3). The molecular structure motif showed a five-fold coordinated Al(III) centre formed by bidentate coordination of two enaminone units and a hydride ligand. the structural identify of **3** in solution was confirmed by ^27^Al NMR spectra that was in accordance to the ligand arrangement and coordination number revealed by X-ray diffraction analysis. Veith et al has also reported on similar problems encountered in the crystal structure elucidation of aluminium hydrido alkoxides [34]. 

The compound **4a** crystallizes in the monoclinic space group P2_1_/n (Table 1) and showed a dimeric structure formed by bridging enaminone ligands (Figure 4). Interestingly, the bridging occurred through the nitrogen atom of the ligand instead of the oxygen atom that has lower lewis basicity due to the vicinity of highly electronegative -CF_3_ group. Similar dimeric compounds containing a central Al2X2-four-membered ring have been reported for aluminium hydrido alkoxides [ROAlH_2_]_2_, hydrido halide alkoxides [ROAlHX] and halide alkoxide [ROAlX_2_]. The bond lengths and angles in **4a** (Table 2) are comparable with those reported for known compounds possessing Al-N, Al-O and Al-H bonds [33]. The Al-H bond length for Al(1)-H(1) and Al(2)-H(2) were found to be 1.493 Å, which compares well with the range (Al − H_terminal_ = 1.47 ± 3 Å) reported for terminal Al-H bonds and shorter than reported values (Al − H_axial_ = 1.61 ± 3 Å, Al − H_equitorial_ = 1.92 ± 3 Å) of bridging H-Al-H bonds, which is known for aluminium alkoxides [35,36]. While the C2-C3 bond distance of 1.330(1) Å represents a double bond character, theC3-C4 bond with a bond length of 1.507(1) Å is in agreement with C-C single bond that confirmed of the hydride migration to the C4 position [37].

The C4-N1 [(1.520(1)Å] and C4-C3 [(1.507(1)Å] bond lengths are longer than a typical delocalized π–bonding system suggesting a more localized electron density in the enaminone backbone. The delocalization of π–electron density in the chelate is apparently prevented by the sp3 hybridized nature of the carbon atom (C4). The effects on the bond lengths of Al-N and Al-O can be attributed to the electronegativity of the ligand at the aluminium centre as witnessed in similar compounds, for example Me_3_AlNMe_3_, H_3_AlNMe_3_ and Cl_3_AlNMe_3_, where a higher electronegative ligand was found to decrease the Al-N bond length. The Al1-O1 and Al1-N1 bond lengths in complex are 1.7520(11)Å, 1.9548(11)Å respectively the electronegative effect affecting the Al2-N2 bond [38]. The Al-N bond length is significantly longer due to the increased steric bulk at the N atoms, which forces them to bind farther from the Al atom and forces the O atoms to bind closer to the Al atom in order to supply it with enough electron density.

The bond angles around the aluminium centre showed a distorted tetrahedral geometry (Figure 4) with two nitrogen atoms, an oxygen atom and a hydrogen atom with bond angles ranging from 104.22(4)° (N1-Al1-O1), 89.94 (3)° (N2-Al1-N1), 114.42(4)° (N2-Al1-O1) and 112.4(6)° (O1-Al1-H1) which are in agreement with the reported values [39].

The intramolecular hydride transfer in 4a might have been induced by the activation of a strongly electron deficient C=C group adjacent to the tert-amino group (*“tert-amino effect”*) present in the enaminone ligand [40,41].

#### 2.1.2. Thermal Behavior of Compounds **3** and **4a**

The decomposition behaviour of precursors **3** and **4a** was studied by thermogravimetric analysis up to 1000 °C (Figure 5). Prior to measurements, the samples were prepared and properly sealed in the glovebox before being introduced carefully into the machine under Nitrogen shower, and subsequently measured under nitrogen. Compound **3** showed a first weight loss of 5% at 115 °C, corresponding to the loss of one hydride ligand by hydrogen desorption. Hydrogen desorption of several organo-hydrido compounds have been reported within this temperature range [42]. Further decomposition step involves the loss of ligand-aluminium bond and loss of free ligand respectively between temperature range of 200–300 °C and thus has promising CVD precursor characteristics at this temperature. The residue mass at 33% could be seen to be slightly higher than the theoretical value of 24%. This deviation could be attributed to carbon or fluoride impurities present in the ligand. Compound **4a** showed loss of hydrogen similar to compound **3**, but apparently evaporated at elevated temperature giving rise to 60% weight loss. This massive weight loss can be attributed to high degree of hydrides and fluorination which massively enhanced its volatility. This also indicated the challenges associated with high volatility that can influence the growth rate and materials yield due to intact transport of precursor to the cold trap. Further heating of the compound resulted in the removal of volatile compounds, yielding a residual mass of 35%, which was unusually lesser than the theoretical mass of 45% due to massive evaporation and high volatility. The powder XRD of the obtained residue revealed amorphous phases of Al_2_O_3_ (Appendix A).

### 2.2. Thin Film Deposition

Compound **4a** was used as a precursor to deposit Al_2_O_3_ film in a low pressure, cold wall CVD chamber with a constant background pressure of 10^−6^ mbar. Prior to loading the silicon wafer into the deposition chamber, it was ultrasonically treated for 10 min in diluted hydrochloric acid, demineralized water, and acetone. Amorphous Al_2_O_3_ films were deposited by maintaining the substrate temperature in the range of 450–500 °C with precursor reservoir maintained at 90 °C as expected of films grown in this temperature range and subsequently annealed under air (1000 °C, 5 h) [43]; to obtain a phase transformed nanocrystalline corundum phase (α-Al_2_O_3_) as confirmed by powder XRD analysis (Figure 6).The as-deposited film showed partial crystallinity, little or no diffraction peaks which is expected for an amorphous substance. Subsequently, the air annealed film showed well defined diffraction peaks matching to thermodynamically stable corundum phase that is normally expected for Al_2_O_3_ above 1000 °C.

The morphology of the films was studied using scanning electron microscopy (SEM) as shown in Figure 7. The obtained micrographs of the as-deposited films revealed a typical densely packed grains of several tens with average diameter of ~194 nm and thickness of ~850 nm (Figure 7). Films grown showed good coverage, homogenously dispersed and adhered to the substrate which is typical for most Al_2_O_3_ reported [44]. The AFM image shows rms value 4.08 nm which clearly depicts the uniformity of the film which corroborates with the SEM image. Annealing of the film led to morphological changes and a plateau-type of structure with diameter of ~430 nm (Appendix A) [45,46]. 

## 3. Materials and Methods

All manipulations were performed under nitrogen atmosphere in the glove box (argon box). Solvents were dried with conventional methods and stored under N_2_ atmosphere with molecular sieves in Schlenk flasks. AlCl_3_ was purified by sublimation before use. [AlH_3_(THF)] was prepared according to literature [47]. TFB-TBA ligand was vacuum dried overnight prior to use. All other reagents were procured commercially from Aldrich and used without further purification. NMR spectra were recorded using a Bruker Avance II 300 spectrometer equipped with a double resonance (BBFO) 5 mm observe probehead with z-gradient coil (Bruker, Rheinhausen, Germany), operating at ^1^H: 300.1 MHz; and ^19^F: 282.4 MHz; or Bruker AV 400, chemical shifts are given in part per million (ppm) relative to TMS (^1^H), CFCl_3_ (^19^F) and Al(H2O)_6_^3+^ (^27^Al) and coupling constants (J) are given in Hz. All 2D NMR experiments were performed using standard pulse sequences from the Bruker pulse program library. NMR spectra were plotted using the Bruker TopSpin 3.2 software.

Elemental analyses were carried out on a HEKAtech CHNS Euro EA 3000. The sample preparation was done inside the glovebox prior to measurement. Deviations of the CHNS data from the calculated values could be attributed to the extraordinary sensitivity of the compounds. TG analysis was performed by a TG/DSC1 (Mettler Toledo GmbH, Germany) apparatus using nitrogen gas and a heating rate of 25 °C/min. Powder X-ray diffraction was measured on a STOE diffractometer with STADI MP system and either Mo Kα (λ = 0.71 Å) or Cu Kα radiation (λ = 1.54 Å). FT-IR spectra were measured using Platinum ATR Spectrometer places inside an argon glovebox with samples analyzed using OPUS software.

The MOCVD tests were carried out in a cold-wall CVD reactor with cooling traps, an internal pressure sensor, and external temperature sensors at low-pressure settings (p 106 bar). Previous works from our working group provide more thorough information [23].

Powder X-ray diffraction (XRD) at room temperature was performed on a STOE-STADI MP diffractometer in the reflection mode with CuKα (1.5406 Å) radiation. The samples’ microstructures were examined using field-emission scanning electron microscopy (FESEM, FEI Nova NanoSEM 430). 

The data collection for X-ray structure elucidation was performed on a STOE IPDS II diffractometer (Mo Kα = 0.71073 Å, 50 kV, 30 mA), and the used programs for structure solution as well as the refinement are SIR-92 [48]; SHELXS [49]; SHELXL [50]; and WinGX [51]. These data are provided free of charge by the joint Cambridge Crystallographic Data Centre or via http://www.ccdc.cam.ac.uk/data_request/cif (accessed on 8 January 2023). Deposition Number 2181692 (complex **4a**) contains the Appendix A for this paper.

### 3.1. Synthesis of Ligand and Complexes

#### 3.1.1. Synthesis of **1**

1-Ethoxy- 4,4,4-trifluorobut-1-en-3-on (ETFB) **1** was prepared by dissolving 2.9 mL (30 mmol, 1.0 equiv.) of ethyl vinyl ether and 12.1 mL (150 mmol, 5.0 equiv.) of pyridine in 80 mL of DCM. This solution was cooled to 0 °C under stirring. Over the course of 30 min, 4.3 mL (30 mmol, 3.0 equiv.) of trifluoroacetic acid anhydride was added dropwise. This solution was left to stir overnight at rt. Afterwards, 300 mL of 3% Na_2_CO_3_(aq) was added. The mixture was extracted with EtOAc (4 × 200 mL). The organic phase was washed with 4 × 100 mL brine and the solvent was removed under reduced pressure. The crude product was purified by vacuum distillation (40 °C, 10^−3^ mbar) and was obtained as a yellow oil that was used for further synthesis. Yield: 69% (3.48 g). 

Characterization Data of ETFB **1**. ^1^H NMR (300.1 MHz, 298 K, CDCl_3_) δ [ppm]: 7.84 (d, ^3^*J*_H,H_ = 12 Hz ^1^H, vinylic CH), 5.80 (d, ^3^*J*_H,H_ = 12 Hz, ^1^H, vinylic CH), 4.06 (q, 2H, CH_2_), 1.39 (t, 3H, CH_3_). ^19^F NMR (282.4 MHz, 298 K, CDCl3) δ [ppm]: −78.8 (s, CF_3_). Elemental analysis calc. for C_6_H_7_F_3_O_2_: C 42.87, H 4.20; found: C 42.47, H 3.90.

#### 3.1.2. Synthesis of **2**

N-(4,4,4-trifluorobut-1-en-3-on)-tert-butylamine (H-TFB-TBA) **2** was prepared by dissolving 5.04 g (30 mmol, 1.0 equiv.) of *ETFB* in 80 mL of toluene and was cooled to 0 °C under stirring. Over the course of 30 min, 2.4 mL (30 mmol, 1.0 equiv.) of tert-butylamine was added dropwise. This solution was left to stir for 1 h. A solid was crystallized from the reaction mixture, washed with n-pentane and recrystallized in a 1:1 toluene/n-pentane mixture, yielding a colorless solid, **2**. Yield 61% (3.57 g).

Characterization Data of H-TFB-TBA **2.**
^1^H NMR (300.1 MHz, 298 K, CDCl3) δ [ppm]: 10.66 (s, ^1^H, NH); 7.26 (q, ^3^*J*_H,H_ = 7.1 Hz, ^1^H, vinylic CH); 5.36 (d, ^3^*J*_H,H_ = 7.1 Hz, ^1^H, vinylic CH); 1.34 (s, 9H, CH_3_).^19^F NMR (282.4 MHz, 298 K, CDCl_3_) δ [ppm]: −76.9 (s, CF_3_). Elemental analysis calc. for C_8_H_12_F_3_NO: C 49.23, H 6.20, N 7.18; found: C 49.54, H 6.44, N 7.02.

#### 3.1.3. Synthesis of **3**, **4**, **4a**, **4b** and **4b′**

HAl(TFB-TBA)_2_
**3**. 

For the synthesis of [HAl(TFB-TBA)_2_] **3**, AlH_3_ was firstly prepared and was used in situ for further synthesis of **3**. In a typical reaction, 0.2 g of AlCl_3_ (1.5 mmol) was dissolved in THF (15 mL) and was slowly added to a slurry of continuously stirred 0.17 g of LiAlH_4_ (4.5 mmol) in THF (20 mL). After stirring for 1 h, two equivalent (2.34 g) of H-TFB-TBA (12 mmol) was added dropwise with evident hydrogen evolution. The slurry was further stirred for 24 h under room temperature. Volatile solvents were removed under vacuum, and the resulting product extracted from THF/n-heptane solution to obtain a white powder and further crystallized in the glovebox using toluene under room temperature to obtain colourless crystals of **3**. Several attempts to obtain quality crystals via several crystallization methods proved unproductive. Yield: 83.6% (2.09 g, 5 mmol).

Characterization Data of HAl(TFB-TBA)_2_
**3**. ^1^H NMR (300.1 MHz, 298 K, C_6_D_6_) δ [ppm]: 7.17 (d, ^3^*J*_H,H_ = 5.74 Hz, ^1^H, 3-H); 5.35 (d, ^3^*J*_H,H_ = 5.87 Hz, 1H, 4-H); 1.30 (s, 9H, 5-H). 

^19^F NMR (282.4 MHz, 298 K, CDCl_3_) δ [ppm]: −76.47 (s, CF_3_). ^27^Al NMR (500 MHz, 298 K C_6_D_6_) δ [ppm]: 66.44 (∆_1/2_ = 6475 Hz). Elemental analysis (C_16_H_23_AlF_6_N_2_O_2_); calculated: C 46.16, H 5.57, N 6.73; found: C 46.04, H 5.43, N 6.96. 

[H_2_Al(TFB-TBA)]_2_
**4**, **4a**, **4b** and **4b′**. 

The synthetic procedure was similar to that described in case of compound (**1**). However, the ligand used was one equivalent instead of two i.e., 1.17 g (6 mmol) to obtain yield of 85% (1.14 g, 2.55 mmol). The product (white powder) was further recrystallized in the glovebox to obtain colourless crystals of (**4a**) at room temperature in a mixture of toluene and THF solution. 

Characterization Data of [H_2_Al(TFB-TBA)]_2_
**4b** and **4b′**. ^1^H NMR (300 MHz, C_6_D_6_) δ 5.09 (dd, *J* = 6.8, 3.5 Hz, ^1^H, 4-H), 4.98 (dd, *J* = 6.7, 3.4 Hz, ^1^H, 4-H′), 3.82 (dp, *J* = 17.7, 3.0 Hz, ^1^H, H-3a′), 3.56 (dt, *J* = 17.0, 3.5 Hz, ^1^H, H-3a), 2.74 (dd, *J* = 17.8, 6.7 Hz, ^1^H, H-3b′), 2.62 (dd, *J* = 17.1, 6.8 Hz, ^1^H, H-3b). ^13^C NMR (101 MHz, C_6_D_6_) δ 145.35 (C2), 145.21(C2′), 122.10 (C1), 119.4(C1′), 105(C4), 104.7(C4′), 56.6(C5) 57.08(C5′), 39.54(C3) 39.27(C3′), 28.98(C6), 29.70(C6′). ^19^F NMR (376 MHz, 298 K, CDCl_3_) δ [ppm]: −73.84, −73.99 (s, CF_3_). ^27^Al NMR (500 MHz, 298 K C_6_D_6_) δ [ppm]: 83.95, 191.16 (∆_1/2_ = 4844 Hz). Elemental analysis (C_16_H_23_AlF_6_N_2_O_2_); calculated: C 43.06, H 5.87, N 6.28; found: C 43.11, H 5.90, N 6.32. 

## 4. Conclusions

This precursor concept based on the unification of both high reactivity (hydride ligands) and stability (Figure 1) demonstrates a plausible pathway to design volatile (low molecular weight) and reactive (low activation energy) precursor compound as potential catalyst for intramolecular C-H activation and functionalization in organo-catalysis. Therefore, it seems reasonable that **4a** will find applications in a range of other transition-metal free catalysis and intramolecular organo transformation (alkene, alkyne and hydroboration reactions). Further investigation of the hydride shift sequence by isotopic labelling using LiAlD_4_ as substrate is currenting ongoing in our laboratory.

## Data Availability

All data presented is available in this manuscript.

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
