# Peer review of "Undirected C-H Bond Activation in Aluminium Hydrido Enaminonates"

_molecules, 2023, doi:10.3390/molecules28052137_

Round 1

Reviewer 1 Report

In this manuscript, Mathur et al. report two new aluminium hydrido complexes and their rapid C-H bond activation and C-C bond formation. In general, the work has been carried out well and is likely to interest the broad readership of Molecules. I, therefore, recommend publication after the following points have been addressed.

1. Scheme 2 has been repeated twice in lines 42 and 46 (page 2).

2. The hydride shift from the H-Al-H to the C(sp2)-H bond is interesting. I think the oxygen atom has more affinity to Al which helps the hydride migration to produce a more stable compound 4a. It would be nice if the authors could add an explanation for the hydride shift in the revised manuscript.

3. Is there any interaction between Al-Al in a solid state of 4a (in the crystal structure)?

4. The TGA, thin film deposition, SEM, and AFM parts are well analyzed and drafted.

5. Is it possible to add some recent years' references? 

Author Response

  1. Scheme 2 has been repeated twice in lines 42 and 46 (page 2).

The title of the schemes rearranged during MS upload, it has been corrected in the MS.

  1. The hydride shift from the H-Al-H to the C(sp2)-H bond is interesting. I think the oxygen atom has more affinity to Al which helps the hydride migration to produce a more stable compound 4a. It would be nice if the authors could add an explanation for the hydride shift in the revised manuscript.

The intramolecular hydride transfer in 4a might have been induced by the activation of a strongly electron deficient C=C group adjacent to the tert-amino group (´´tert-amino effect´´) present in the enaminone ligand. The authors have also added this explanation for the hydride shift in the MS.

  1. Is there any interaction between Al-Al in a solid state of 4a (in the crystal structure)?

We did not find such interaction between Al-Al while solving the crystal structure of 4a.

  1. The TGA, thin film deposition, SEM, and AFM parts are well analyzed and drafted.
  2. Is it possible to add some recent years' references? 

The authors have also added the references as requested.

Reviewer 2 Report

This article provides a detailed review of the synthesis and characterization of two new aluminium hydride complexes. The authors have provided an in-depth analysis of the structural motif of both the mono- and di-hydrido-aluminium enaminonates which was confirmed by single crystal structural data. The intramolecular hydride shift involving the migration of a hydride ligand from aluminium centre to the alkenyl carbon of the enaminone ligand was also probed and verified. Therefore, I recommend this article for publication after minor revisions.
-The very wide signal you are seeing may be due to the glass of the NMR tube and its background signal is 27Al. This problem can be avoided in several ways:
1. You can use a quartz tube to record NMR spectra.
2. You can subtract the signal by increasing the Prescan Do Delay (DE).
3. Record a separate experiment with the same number of scans and receiver gain and subtract it from the spectrum of the corresponding sample [10.3390/molecules28041518].
-The authors must explain how their work is similar and different to the literature of the problem. They should make it clear what the new results of their work are and how they build upon the previous work in the field. For instance, what are the 27Al NMR signals of montmorillonite when intercalated with large Al- and Al/Ce-polyhydroxocomplexes? Are the signals similar to those of the compounds in 10.1134/S1995078015050031?
-It is essential that the authors provide a detailed description of the parameters of the NMR spectrometer equipment and the pulse sequence used for NMR experiments, including mix time, transients, pulse width, power, points, increments, sweep width, and relaxation delay, in order to make the paper suitable for publication.

Author Response

Authors comments for the second reviewer.

Undirected C-H Bond Activation in Aluminium Hydrido Enaminonates

  • The very wide signal you are seeing may be due to the glass of the NMR tube and its background signal is 27Al. This problem can be avoided in several ways:
    You can use a quartz tube to record NMR spectra.
    2. You can subtract the signal by increasing the Prescan Do Delay (DE).
    3. Record a separate experiment with the same number of scans and receiver gain and subtract it from the spectrum of the corresponding sample [10.3390/molecules28041518].

Response:

Thank you for this very important observation, unfortunately, we don’t have the possibility or options to measure with quartz tubes in our institute and hence, NMR measurements were only carried out using specially constructed glass tubes for inert NMR measurements, However, we could not repeat the measurements as suggested because 27Al NMR measurement is currently out of order in our institute. The authors were able to remove the background noise using zero filling and LP option in topspin software (see figure in SI).

  • The authors must explain how their work is similar and different to the literature of the problem. They should make it clear what the new results of their work are and how they build upon the previous work in the field. For instance, what are the 27Al NMR signals of montmorillonite when intercalated with large Al- and Al/Ce-polyhydroxocomplexes? Are the signals similar to those of the compounds in 10.1134/S1995078015050031?

Response: Thank you for pointing out this important points, hydride migration in Al-H compounds reported in literature involved intermolecular shift via stoichiometric reactions with alkenes and alkynes, activation of unsaturated substrates by aluminium hydrides, activation of molecular chalcogenides etc (as seen in the review 10.1016/j.ccr.2017.03.017), whereas we reported an undirect intramolecular C-H bond activation by intramolecular hydride shift in Al-H compound, making our work unique, this has also been highlighted in the manuscript. The 27Al NMR of the compounds were comparable with literature known Al-H compounds for both compounds as referenced in the MS during the 27Al descriptions (please see line 127-137 in MS). The authors only limited the 27Al NMR study to 27Al NMR spectrum of previously studied Al-H compounds.

  • It is essential that the authors provide a detailed description of the parameters of the NMR spectrometer equipment and the pulse sequence used for NMR experiments, including mix time, transients, pulse width, power, points, increments, sweep width, and relaxation delay, in order to make the paper suitable for publication.

Response: Thank you for the correction, the NMR spectra were recorded using a Bruker Avance II 300 spectrometer equipped with a double resonance (BBFO) 5 mm observe probehead with z-gradient coil (Bruker, Rheinhausen, Germany), operating at 1H: 300.1 MHz; and 19F: 282.4 MHz; or Bruker AV 400, chemical shifts are given in part per million (ppm) relative to TMS (1H), CFCl3 (19F) and Al(H2O)63+ (27Al) and coupling constants (J) are given in Hz. All 2D NMR experiments were performed using standard pulse sequences from the Bruker pulse program library, this has been clearly highlighted in the MS as you suggested (please see line 256-263).
